# Mycotransformation of Commercial Grade Cypermethrin Dispersion by *Aspergillus terreus* PDB-B Strain Isolated from Lake Sediments of Kulamangalam, Madurai

**DOI:** 10.3390/molecules29071446

**Published:** 2024-03-23

**Authors:** Priyadharshini Kannan, Hidayah Baskaran, Jemima Balaselvi Juliana Selvaraj, Agnieszka Saeid, Jennifer Michellin Kiruba Nester

**Affiliations:** 1Department of Microbiology, The American College, Madurai 625002, Tamil Nadu, India; priyadharshinimic31@gmail.com (P.K.); hidayahb1608@gmail.com (H.B.); 2Department of Chemistry, The American College, Madurai 625002, Tamil Nadu, India; jemimamanojjaden@gmail.com; 3Department of Chemical Engineering, Politechnika Wroclawska, 50-370 Wroclaw, Poland; agnieszka.saeid@pwr.edu.pl

**Keywords:** cypermethrin, mycotransformation, pesticide, fungi, microcosm

## Abstract

A fungal isolate *Aspergillus terreus* PDB-B (accession number: MT774567.1), which could tolerate up to 500 mg/L of cypermethrin, was isolated from the lake sediments of Kulamangalam tropical lake, Madurai, and identified by internal transcribed spacer (ITS) sequencing followed by phylogenetic analysis. The biotransformation potential of the strain was compared with five other strains (A, J, UN2, M1 and SM108) as a consortium, which were tentatively identified as *Aspergillus glaucus*, *Aspergillus niger*, *Aspergillus flavus*, *Aspergillus terreus*, and *Aspergillus flavus*, respectively. Batch culture and soil microcosm studies were conducted to explore biotransformation using plate-based enzymatic screening and GC-MS. A mycotransformation pathway was predicted based on a comparative analysis of the transformation products (TPs) obtained. The cytotoxicity assay revealed that the presence of (3-methylphenyl) methanol and isopropyl ether could be relevant to the high rate of lethality.

## 1. Introduction

The increasing population and decreasing agricultural spaces across the globe put enormous pressure on current agricultural practices to increase their yield and meet the demands of food sustainability. The food supply of important grain crops such as rice, wheat and maize are heavily affected by a 5 to 20% loss of crop yield due to insect consumption [1]. Such scenarios often justify the use of pesticides that can substantially increase crop yield. Even though their use aids in supplying humans with food, they also contribute to widespread organic pollutants in soil, surface water, and groundwater (via leaching or runoff) and have an impact on biodiversity through biomagnification and bioaccumulation. Nearly 25 million square kilometers of land are at risk of being polluted by more than one type of pesticide, according to a recent estimate of the global risk of pesticide pollution [2]. These areas include biodiversity hotspots, regions with limited water supplies, and low- to middle-income countries.

Even the terminology “pesticide” is all-encompassing of various types such as herbicides, fungicides, and insecticides. The most widely used pesticides around the world are herbicides at nearly 50%, followed by insecticides, and others. Overexploitation and improper storage often cause an increase in their residual concentrations in nearby water resources, such as tropical lakes and wells, used for irrigation purposes. Based on their chemical makeup, insecticides can generally be divided into several groups, such as pyrethroids, carbamates, organochlorines, and organophosphates. These synthetic pesticides are employed more frequently than insecticides derived from naturally occurring plants or mineral oils [3]. The chemicals resmethrin, permethrin, tetramethrin, deltamethrin, cypermethrin, and -cyhalothrin are all components of synthetic pyrethroid insecticides.

### 1.1. Cypermethrin as a Pollutant

Cypermethrin (CYP) is a fourth generation pyrethroid ester insecticide extracted from *Chrysanthemum cinerariaefolium*, widely used in India, and is a component of most household insecticides against mosquitoes and pet ectoparasites [4]. Although considered a safer alternative to organophosphorus pesticides in low dosages, the excessive and indiscriminate use of these pesticides often leads to bioaccumulation through the food chain. Residual cypermethrin in the food chain has been shown to be toxic to fish and mice models and has been linked to several cancers of the lung, brain, bladder, ovary, hormonal imbalance, and Parkinson’s disease (PD) [5]. Furthermore, the Environmental Protection Agency (EPA) lists cypermethrin as a possible human carcinogen, implicating the disastrous effects it can have on human and animal health in inordinate residual concentrations. As a result of the persistent nature of the molecule in runoff water and bioaccumulation in water-based ecosystems, several detrimental effects have often been observed in aquatic organisms, especially fish. Erratic swimming and migration behavior, disorders during fish development stages and reproduction, significant alterations in biochemical profiles such as endocrine malfunctions, genotoxicity, neurotoxicity, and observation of oxidative stress injury are a wide range of potential complications observed in aquatic organisms [6].

### 1.2. Mycobiodegradation as a Solution

Several studies have found using microorganisms as a single inoculant or a consortium for the transformation of toxic and hazardous organic or inorganic pollutants into useful or harmless compounds in situ or ex situ as a competent tool. The fundamental trait of fungi as saprobic decomposers and their association with biogeochemical cycling processes in the soil and litter zone are additional advantages for their utility in the microbial biodegradation process. The capability of microbial esterase enzymes that selectively target the hydrolysis of pyrethroid compounds is discussed in a recent study [7]. Many fungi, including those that cause wood rot, have been shown to produce a wide range of enzymes, including laccases, peroxidases, esterases, and other hydrolases with enormous industrial uses. Their wide range of carbon sources, including aliphatic hydrocarbons, and special capacity to co-metabolize several xenobiotics enable them to withstand dangerous metal concentrations [8].

### 1.3. Current Studies

There is no dearth of literature that enlists the potential of microorganisms such as bacteria and fungi to degrade potentially toxic organic pollutants into harmless metabolites [9]. Although studies have been reported on the biodegradation of residual cypermethrin in terrestrial environments by bacteria such as *Lactobacillus* sp., *Bacillus subtilis*, *Bacillus cereus*, *Streptomyces toxytricini*, *Pseudomonas* sp., *Rhodococcus* sp. and *Serratia* sp., the literature on mycoremediation has been limited and restricted to *Eurotium cristatum*, *Fusarium* sp., and *Aspergillus* sp. [4,10,11,12,13,14] (Table 1). However, most biodegradation tests report the presence of the cypermethrin metabolite 3-phenoxybenzoic acid (3-PBA), which is again a recalcitrant compound that poses various health risks in the food chain.

This study attempts to determine the biodegradation capacity of the fungal strain and a fungal consortium isolated from the lake sediments of a tropical lake in Madurai, Tamil Nadu. The Kulamangalam lake is a major source of water for irrigation purposes in the city of Madurai and is fed by perennial rainwater. 

## 2. Results and Discussion

### 2.1. Physicochemical Properties of Soil

The soil was identified to consist predominantly of clay and loam and has a pH in the range of mild to strongly alkaline. The EC value for the soil was 1.8 dSm^−1^, indicating its very slightly saline nature and is generally considered conducive to irrigation and agricultural purposes. Calcareousness, the available and total nitrogen in soils, is low, whereas available phosphorus and potassium are high. The organic carbon available for metabolism, and the iron, manganese, and copper available in the sample were determined to be sufficient for cultivation, except for the available zinc (refer to Appendix A).

### 2.2. Isolation and Identification of Fungi

Sixteen fungal isolates were obtained from lake sediment samples and were tentatively identified according to colony morphology and spore arrangement (Appendix A). The fungal isolate that exhibited the maximum potential for cypermethrin degradation was identified by internal transcribed spacer (ITS) sequencing followed by phylogenetic analysis (Figure 1). The sequence obtained was homologous to *Aspergillus terreus* ATCC type culture 1012 (NR_131276.1) and was submitted to the Genbank database (accession number: MT774567.1). In addition to the *Aspergillus terreus* PDB-B strain, five other strains (A, J, UN2, M1, and SM108) that were included in the consortium biodegradation studies were tentatively identified as *Aspergillus glaucus*, *Aspergillus niger*, *Aspergillus flavus*, *Aspergillus terreus*, *Aspergillus flavus*, respectively.

### 2.3. Screening for Cypermethrin Resistance

Of the 16 fungal cultures isolated from lake sediments, five fungal strains (A, J, UN2, M1 and S + M10^8^) were able to tolerate 500 mg/L of cypermethrin in potato dextrose agar plates and used cypermethrin as the sole carbon source in minimal salt medium. The B isolate exhibited more confluent growth in both mediums compared to other fungal strains and was used for further studies. In plate-based screening assays, a higher degree of biodegradation was observed upon co-inoculation of the fungal strains along with PDB-B, in terms of halo zone formation. During single strain inoculation, PDB-B showed the highest degree of halo zone formation. Therefore, the effect of mono-inoculant versus consortia was studied in flask-based and soil microcosm studies.

### 2.4. Screening for Extracellular Enzymes Involved in Cypermethrin Degradation

After screening the fungal strains for extracellular enzyme production (Refer Appendix A), the fungal strains M1 (*Aspergillus terreus*), UN 3 (*Aspergillus niger*), H3 (*Aspergillus glaucus*), I (*Aspergillus niger*), and B were found to produce β-glucosidase. The fungal strain M6 (*Aspergillus nidulans*) was identified to produce urease enzyme. The fungal strain I (*Aspergillus niger*) produced esterase enzyme in the plate-based screening. Fungal strain M8 produced both urease as well as β-glucosidase enzymes.

Appendix A shows the screening results for the production of lignin-degrading enzymes. Fungal strains D (*Aspergillus niger*), H1 (*Aspergillus glaucus*), I (*Aspergillus niger*), J (*Aspergillus niger*), UN3 (*Aspergillus niger*), and S + M 10^8^ (*Aspergillus flavus*) produced a clearance zone in the bromophenol plate-based assays, indicating laccase production. However, only strain M6 (*Aspergillus nidulans*) exhibited brown coloration on plates supplemented with guaiacol as a laccase substrate. Fungal isolates B, H1 (*Aspergillus glaucus*), M1 (*Aspergillus terreus*) and M8 can produce lignin peroxidase. Manganese peroxidase was excreted by strains I (*Aspergillus niger*), M1 (*Aspergillus terreus*), and UN2 (*Aspergillus flavus*). All strains, except M7 and M8, could utilize lignin as the sole carbon source and exhibited growth in minimal salt medium supplemented with lignin. Similarly, all fungal isolates except H (*Aspergillus glaucus*), I (*Aspergillus niger*), and J (*Aspergillus niger*), turned positive for the screening for the secretion of the polyphenoloxidase enzyme. On the other hand, only the isolates A (*Eurotium herbariorum*), I (*Aspergillus niger*), J (*Aspergillus niger*), UN2 (*Aspergillus flavus*), UN3 (*Aspergillus niger*), S + M 10^8^ (*Aspergillus flavus*) and M8 exhibited clearance zones when plated on plates containing lignin mimicking dyes.

Appendix A show the presence and absence of clearance zones around the fungal colony. It indicates the ability of the fungal strain to produce the respective enzyme, and the ability of the fungal isolates to utilize the respective lignin substrate and exhibit growth.

In the case of alkali soils, they are often found to have very low levels of organic carbon and nitrogen (Refer Appendix A). This ultimately causes low levels of urease and dehydrogenase activity in soils [24]. Under these conditions, *Aspergillus nidulans* was able to utilize urea, indirectly implying the presence of urea in the soil (Appendix A) [25]. 

β-glucosidases produced by microorganisms, are enzymes that can play diverse roles in biomass conversion, glycolipid breakdown, and lignification, biocontrol, induction of phytohormones, plant and insect cell wall catabolism. These functionalities of this enzyme family indicate the diverse biotechnological applications potential in industries such as food, surfactant, biofuel, and agriculture [26]. There are numerous reports of β-glucosidase being secreted, isolated, and purified from *Aspergillus* strains. For instance, Yan, 2016, an *Aspergillus terreus* strain was identified to produce an extracellular β-glucosidase (BGL) that could hydrolyse soybean isoflavone [27]. When a selected strain of *Aspergillus niger* was cultivated in solid state fermentation (SSF), it was observed to produce a thermostable BGL that was partially purified and characterized [28]. Similarly, an *A. niger* isolate from effluent contaminated soil of a cotton ginning mill was found to produce a β-glucosidase enzyme [29]. A β-1,4-glucosidase (BG) was purified from *Aspergillus glaucus* fermentation liquor [30]. Similarly, in our study, we have isolated fungal strains M3 10^1^ (*Aspergillus terreus*), UN 3 (*Aspergillus niger*), H3 (*Aspergillus glaucus*), I (*Aspergillus niger*), and B that were found to produce β-glucosidase. We propose that these strains could be further tested for industrial applications. 

Esterases (carboxylesterases, EC 3.1.1.1) and lipases, collectively known as lipolytic enzymes, play a crucial role in catalyzing the cleavage and formation of ester bonds. These enzymes have been extensively studied, isolated, purified, and characterized from diverse sources including animal and plant tissues, as well as microorganisms. Their primary function involves facilitating transesterification, esterification, and enantioselective hydrolysis reactions of ester molecules with a carbon chain length exceeding 10 atoms or short-chain triglycerides. Notably, these enzymes exhibit remarkable enantioselectivity, stability in organic solvents, and a broad substrate specificity for non-natural compounds.

Microbial esterases, in particular, hold significant potential for various applications in biocatalysis, particularly in the production of optically pure compounds such as carboxylic acids, primary, secondary, and tertiary alcohols. In the pharmaceutical industry, they find utility in synthesizing chiral compounds like antidepressants and antibiotics. Additionally, these enzymes contribute to the recycling of by-products generated in the pulp and paper industry, as well as the manufacture of short-chain esters associated with fragrances and flavors for use in detergents and cosmetics. Bacterial strains such as *Bacillus subtilis*, *Rhodococcus* sp. LKE-028, and *B. licheniformis* have been isolated and purified to obtain these valuable esterases for further research and applications [4,7].

In this study, the fungal strain I (*Aspergillus niger*) produced an esterase enzyme in plate-based screening. In previous reports, it was observed that solid state fermentation of oat spelt xylan and sugar beet pulp by *Aspergillus niger* strain produced an inducible ferulic acid esterase (FAE-III) and inducible cinnamoyl esterase, respectively, which was partially characterized [31,32]. Similarly, one of the fungal genes that encoded for FAE from *Aspergillus tubingensis* and *Aspergillus niger* was identified and cloned from *Rhizomucor miehei* (RmEstA) and expressed in *Escherichia coli* for further characterisation [33,34]. Similarly, fungal strain I could also be subjected to further analysis for industrial application. However, fungal esterases seem to be less studied in comparison to bacterial esterases, despite their commercial value, and very few investigations have been reported on fungal esterases than on those obtained from bacterial or mammalian sources.

#### Lignin-Degrading Enzymes

Ligninolytic fungi produce a wide range of extracellular peroxidases and laccases that decomposes recalcitrant organic pollutants present in water or soil. Some of the enzymes that have been quite extensively studied are phenol oxidase (laccase), Mn-dependent peroxidase (MnP), lignin peroxidase (LiP), superoxide dismutase, glucose oxidase, glyoxal oxidase, and aryl alcohol oxidase [35,36]. 

Similarly, in our study, we isolated four strains of *Aspergillus niger* (I, J, D, and UN3) that could produce laccase and can use lignin as the only carbon source for its metabolism in plate-based study. However, strain M3 10^6^ (*Aspergillus nidulans*) alone used guaiacol as a laccase substrate. Laccases are multicopper oxidase enzymes that oxidize phenolic compounds and are encoded by the *yA* gene in *Aspergillus nidulans*. Studies reveal that mutations in the orthologous *yA* gene, which is expressed during sporulation, are responsible for the conversion of green conidia to yellow [37,38]. This was observed in the growth and colony morphology of the *Aspergillus nidulans* strain. We also isolated two strains of *Aspergillus glaucus* (*Eurotium herbariorum*), one of which decolorised Congo Red, a lignin mimicking dye, while the other produced lignin peroxidase. Similarly, laccases exhibit high decolorization ability of dyes that mimic lignin structure such as bromophenol blue, and congo red. LiP is a heme containing enzyme which can breakdown phenolic as well as non-phenolic substrates with higher oxidation potential compared to laccase and MnP [39]. The *Aspergillus glaucus* strain AJAG1 was found to produce lignin peroxidase during fipronil degradation [40]. The fungal strain M 10^7^ *Trichoderma viride* produced a polyphenol oxidase. 

Most of the strains produced a brown zone of oxidation around their colonies in the presence of tannic acid, which is considered an ambiguous indication of production polyphenol oxidases (PPOs). As the name indicates, this family of enzymes consists of mainly two modes of enzymatic action, hydroxylate monophenolic substrates (tyrosine monophenolase activity) and oxidize phenolic derivatives at their ortho positions. Both tyrosinase and catechol oxidase can oxidize phenolic and polyphenolic substrates into precursors for melanin precursors. In Ascomycota, PPOs have been shown to have several implications in enzymatic degradation of pesticides and have been studied through expression in different heterologous hosts such as a *Trichoderma reesei* tyrosinase produced in *Pichia pastoris* strain and expressed in *Trichoderma reesei* strain [41].

### 2.5. Studies on Biodegradation of Cypermethrin

#### 2.5.1. Studies on Change in pH, Dry Mycelial Weight, and Optical Density

The biodegradation capacity of fungal isolate B (PDB-B) and a fungal consortium (C) was determined under two different conditions in liquid medium (denoted as LNM-B, LNM-C), and MSM (MSM-B, MSM-C), by measuring the change in biomass, pH, and optical density. No change was observed in the pH and optical density of the LNM and MSM control samples.

Figure 2 compares the change in dry mycelia between different media, the fungal isolate B and the fungal consortium. It can be observed that maximum dry mycelial weight was observed for fungal isolate B grown in PDB. Figure 3 and Figure 4 show the change in optical density and pH of the fungal medium inoculated with B and the fungal consortium at 500 mg/L of cypermethrin. Again, the fungal isolate B grown in PDB does not exhibit a change in pH beyond 3.5 to 4, compared to the variations observed in the other inoculum.

The initial findings indicate a pronounced increase in biomass for fungal isolate B across both media types, with a peak observed around day 6. This growth phase is interpreted as indicative of active adaptation and proliferation of the fungal isolate within the provided nutrient environments. Subsequent fluctuations in biomass are attributed to a complex interplay between growth-promoting factors and constraints such as nutrient depletion and the accumulation of metabolic by-products, underscoring the dynamic equilibrium between microbial expansion and environmental limitations.

The richer nutrient profile of PDB is hypothesized to facilitate more robust growth compared to MSM, although this is counterbalanced by more pronounced fluctuations post-peak, likely due to accelerated nutrient consumption and waste product accumulation. The introduction of consortium cultures in both media types reveals additional complexities in biomass dynamics, with potential synergistic or antagonistic interactions among the constituent organisms influencing overall growth patterns. These interactions may account for phases of biomass stabilization or abrupt shifts, highlighting the intricate relationships within microbial communities.

The number of fungal spores is quantified by optical density at a wavelength of 700 nm or greater, as indicated in the research conducted by Morris and Nicholls (1978). Optical density (OD) measurements at wavelengths of 700 nm or higher are used to estimate fungal spore concentrations in water suspensions, providing a reliable method for assessing spore concentrations of various fungal species like *Penicillium digitatum*, *Penicillium italicum*, and *Geotrichum candidum.* However, in the case of filamentous fungi, OD reading can be influenced by factors such as the length and branching pattern of hyphae, the presence of conidia (spores), and the aggregation of mycelial masses. These factors can lead to significant variability in OD readings resulting in conflicts with the actual biomass present, and dry biomass values.

#### 2.5.2. Analysis of Degradation Efficiency and Products by GC-MS

Batch studies of fungal isolates and consortium in liquid medium potato dextrose broth and minimal salt medium are indicated as LNM-B, LNM-C, MSM-B, MSM-C, and soil microcosm studies are indicated as Soil-B and Soil-C. The chromatogram of the GC-MS analysis (Figure 5, Figure 6, Figure 7, Figure 8, Figure 9 and Figure 10) of the samples and the control provides a stark difference between the retention times of the cluster of peaks in the control and test samples, indicating the degradation process and the presence of degraded metabolites, and the absence of cypermethrin in all samples confirms biodegradation.

Table 2 and Table 3 present the metabolites found in the control and samples LNM-B, LNM-C, MSM-B, MSM-C, Soil-B, and Soil-C. Key derivatives of cypermethrin such as α-methyl-benzenemethanol, 1-phenyl-ethanone, ethenyl-benzene, 1-phenylethyl ester formic acid, (3-methylphenyl) methyl ester formic acid, (3-methylphenyl) methanol, isopropyl ether, ethyl-benzene, α-hydroxy-3-phenoxy-benzeneacetonitrile, and phenol were found in the samples. It can be observed that α-methyl-benzenemethanol is present in all test samples in large quantities in decreasing order of LNM-B > MSM-B > LNM-C > MSM-C. 1-phenylethyl ester formic acid and (3-methylphenyl) methyl ester formic acid is also observed in all test samples, but not in large quantities, with a percentage area in the range between 0.06 and 0.15. Fungal strain B as well as consortium grown in LNM supplemented with cypermethrin showed the presence of (3-methylphenyl) methanol and isopropyl ether, while these were absent in both MSM media. Similarly, the fungal consortium grown in MSM exhibited the presence of ethenyl benzene, while all other samples displayed fragments of ethylbenzene.

Furthermore, except for the fungal consortium grown in LNM supplemented with cypermethrin, all other samples MSM-C, MSM-B, and LNM-B revealed the presence of 1-phenyl-ethanone, phenol, and 1-phenylethanol, respectively. The Soil-B test sample expresses an extra mass fragment that is phenol. Residual cypermethrin and 3-phenoxybenzaldehyde are present in Soil-B and Soil-C samples, in similar concentrations. 

The degradation pathway of both fungal strain B and the fungal consortium was studied in both potato dextrose broth (LNM), as well as mineral salt medium (MSM), at the end of a 30-day degradation period. The most commonly studied cypermethrin biodegradation pathway begins with ester hydrolysis which is the initial phase in most pyrethroid metabolic pathways [42]. The ester hydrolysis produces carboxylate, 3-(2,2-dichloroethenyl)-2,2-dimethylcyclopropanecarboxylate and cyano-3-phenoxybenzyl alcohol, which contains a cyanohydrin functional group that is unstable in water. HCN is removed from cyanohydrin in a nonenzymatic reaction or by a pyrethroid hydrolase enzyme resulting in the production of 3-phenoxybenzaldehyde (3-PBA) that is the most common microbial byproduct of cypermethrin biodegradation [43]. Then 3-phenoxybenzaldehyde is further broken down into 3,4-dihydroxybenzoate and phenol through the permethrin pathway, and 3-(2,2-dichloroethenyl)-2,2-dimethylcyclopropanecarboxylate is broken down to release CO_2_.

However, in our GC-MS analysis, all test samples showed high concentrations of α-methyl-benzenemethanol (1-Phenylethanol) (refer to Appendix A). GC-MS analysis of LNM-C, MSM-B, and LNM-B revealed the presence of 1-phenylethanol and ethylbenzene. Several fungal strains *Botrytis cinerea*, *Trametes versicolor*, *Mortierella isabelline*, *Cunninghamella echinulata* var. *elegans*, *Cladophialophora* sp., *Helminthosporium*, and *Colletotrichum acutatum* were found to be capable of the conversion of ethylbenzene and several acetophenone derivatives to their respective optically active 1-phenylethanols [44,45]. Certain fungal strains including *C. elegans* and *M. isabellina* were also found to produce 2-phenylethanol as derivatives [46]. Although there is limited literature on metabolic oxidations of cypermethrin to ethylbenezene, it can be suggested that the fungal consortium is capable of oxidizing cypermethrin to ethylbenzene because of the lack of cypermethrin in the test sample. The presence of a high concentration of 1-phenylethanol and sufficient ethylbenzene is indicative of the presence of an enzyme responsible for benzylic hydroxylation. These biotransformations are often propagated by hydroxylases. Moreover, nonspecific fungal peroxygenases selectively shift the peroxide-borne oxygen to the carbon skeleton of diverse substrates, including pesticides. These enzymes can catalyze reactions such as one-electron oxidations, hydroxylations, dealkylations, epoxidations, inorganic halides, and oxidations of organic heteroatoms. The wide array of substrates catalysed by these enzymes and their by-products have been observed to be quite similar to that of classical heme peroxidases and cytochrome P450 monooxygenases [47]. It is our assumption that these enzymes could be involved in the biotransformation reactions of cypermethrin. More studies are needed to decipher enzyme-substrate interactions.

In contrast, in the MSM-C chromatogram, 1-phenylethanol, acetophenone, and ethenylbenzene were found at high concentrations. Fungi have been studied for their enantioselective biological reduction of acetophenone to its derivatives. Live biomass of *Trichothecium* sp. was found to act as a good biocatalyst for reduction of acetphenone derivatives to their corresponding alcohols [48]. Similarly, the fungal isolate *Trichothecium roseum* was also found to produce (R)-1-phenylethanols by the bioconversion of acetophenones [49]. A similar ability of *Aspergillus* species was observed to asymmetrically reduce acetophenone. A fungal strain *A. glaucus* MA0200 reduced acetophenone to phenol [50]. The biodegradation of acetophenone to phenol is catalyzed by acetophenone monooxygenase and alcohol dehydrogenases releasing a pungent flavor. However, the role of different fungal species in the biotransformation of acetophenone derivatives should be further studied to derive the cypermethrin pathway [51]. The bioproduction of enantiomerically pure secondary chiral alcohols such as (R)-1-phenylethanol are industrially relevant as they are biosynthetic precursors for several bioactive natural products such as pharmaceuticals, agrochemicals, hormones, and proteins. Biocatalysis has economic relevance in the circular economic integration of concepts such as biorefinery and waste biomass conversion of agroindustrial waste into bioproducts. This bioreduction property of the fungal consortium could be further studied for industrial applications.

In addition to these, (3-methylphenyl) methanol and isopropyl ether were also found in the LNM-B and LNM-C chromatogram at average compositions. The role of (3-methylphenyl) methanol and isopropyl ether, in the biodegradation pathway of cypermethrin is unknown. Moreover, ethenylbenzene, which is found only in MSM-C, is also called styrene. *Aspergillus niger* CGMCC 0496 has been shown to enantioselectively bio-hydrolyze various substituted styrene oxides [52]. 

Similarly, among all samples, 1-phenylethyl ester formic acid and (3-methylphenyl) methyl ester formic acid were found at minimal concentrations. The role of these compounds in the cypermethrin pathway was unknown until now.

GC-MS analysis of MSM-B revealed the presence of two new compounds, namely phenol and α-hydroxy-3-phenoxy-benzeneacetonitrile. The presence of ethylbenzene and 1-phenylethanol along with phenol could be explained by the biodegradation pathway of permethrin. Oxidations of cypermethrin in the benzene group could result in the presence of phenol, which is one of the end products of the permethrin pathway. These reactions are catalyzed by a set of hydrolases, dehydrogenases, and dioxygenases involved in the permethrin pathway. Phenol is further degraded to catechol by a different set of enzymes, taken through the nitrobenzene pathway. From the nitrobenzene pathway, intermediates are converted to pyruvate and acetaldehyde, for further metabolism, and released as CO_2_. A fungal strain *Cladosporium* sp. HU was isolated from activated hydrolyzed sludge with the ability to biologically convert fenvalerate by the hydrolysis of carboxylester linkage [53]. Similarly, an actinomycetes strain *Streptomyces aureus* HP-S-01, again an isolate from activated sludge, was able to hydrolyse the carboxyl-ester bond to efficiently break down deltamethrin to α-hydroxy-3-phenoxy-benzeneacetonitrile and 3-phenoxybenzaldehyde (3-PBA). α-hydroxy-3-phenoxy-benzeneacetonitrile was also one of the six intermediate products when cyhalothrin was subjected to biotransformation by *B. thuringiensis* ZS-19 through the cleavage of the diaryl bond and ester link [54]. From the literature, we could arrive at the conclusion that the absence of 3-phenoxybenzaldehyde in all samples suggests its rapid degradation by the fungal consortium. Furthermore, the presence of carboxylesterases could be debated because the production of α-hydroxy-3-phenoxy-benzeneacetonitrile necessitates its secretion. The enzymes could be further purified and characterised by further studies. 

From studies in liquid medium and the transformation products (TPs) predicted from the EAWAG-BBD pathway and their SMILES, we effectively conclude that the whole subset of enzymes, including hydrolases, carboxylesterases, dehydrogenases, mono- and peroxygenases, are involved in the pathway of bioreduction and biotransformation to completely mineralize cypermethrin to CO_2_ (Table 4 and Figure 11). 

### 2.6. Soil Microcosm Studies

Contrary to the data observed in the liquid media studies, we found residual cypermethrin in soil microcosms incubated for a treatment period of 30 days. In liquid media, the rate of the biotransformation is influenced by several physicochemical factors such as pH, temperature, inoculum volume of the microbial inoculant, nutrient availability, the enzymatic spectrum of fungal strains, and toxicity of pyrethroid concentration [55,56,57]. However, the effectiveness of pyrethroid biotransformation also depends on the mobility and reach of the soil microorganisms and additional microbial inoculant and may be potentially influenced to a large extent by the pH, temperature, moisture, organic matter content, and soil texture of the soil [58,59]. 

As discussed above, the presence of 3-phenoxybenzaldehyde is propagated by a pyrethroid hydrolase enzyme. This is taken to the permethrin pathway for further degradation. However, it is interesting to note that both the liquid medium sample MSM-B and the soil microcosm sample, Soil-B, included phenol as an end product. The presence of a simpler molecule such as phenol is highly suggestive of a much better degradation rate compared to other samples. Therefore, we are also able to arrive at the conclusion that the presence of phenol indicates that the fungal isolate B is much more efficient in degrading cypermethrin than the fungal consortium.

Given the production of these enzymes, strains within the *Aspergillus nigri* complex, particularly those producing laccase, lignin peroxidase, manganese peroxidase, and esterase, could play significant roles in the biodegradation of cypermethrin. Their enzymatic activities can contribute to various degradation pathways, leading to the breakdown of the pesticide’s complex structure. Strains producing laccase and polyphenoloxidase could initiate the degradation process through oxidation. Those with lignin peroxidase and manganese peroxidase could further degrade aromatic components and other complex structures within cypermethrin. Strains that produce esterase could directly hydrolyze cypermethrin, given its ester linkages. While the roles of β-glucosidase and urease in cypermethrin biodegradation might be indirect, their activity could support the overall metabolic capacity of the microbial community involved in degradation.

#### Brine Shrimp Lethality Assay (BSLP)

The lethality rate of all liquid media samples was tested against the marine vertebrate *Artemia salina*, after 30 days of incubation with 500 mg/L cypermethrin (Table 5). It is evident from the table that the fungal isolate and consortia incubated in mineral salt medium (MSM) show the least lethality (around 30%) when compared to the control sample and PDB medium. The results of the toxicity assay are consistent with the GC-MS analysis. The liquid media LNM-B and –C exhibited higher lethality rates compared to those of MSM-B and –C. We attribute the presence of (3-methylphenyl) methanol and isopropyl ether to the high rate of lethality.

## 3. Materials and Methods

### 3.1. Collection of Samples

Lake sediment samples were collected in sterile zip lock bags at the following coordinates 10.016699, 78.1125677 near Kulamangalam village in the north Madurai taluk of Madurai district, Tamil Nadu, India, and transported to the laboratory for further processing. Soil samples were sieved, and large particles such as stones and plants were removed to isolate the fungi. 

### 3.2. Determination of the Physicochemical Properties of Soil

Soil samples were subjected to analysis of the following physiochemical properties characterized by the following properties: pH, electroconductivity, organic matter, and available NPK according to standard protocols [60]. 

### 3.3. Isolation and Identification of Fungi

A volume of 0.1 mL of the inoculum obtained from dilutions (10^−3^, 10^−4^, and 10^−5^) of the soil sample was aseptically spread on agar plates containing potato dextrose medium (PDA) and incubated at room temperature for 48 h. Axial cultures of the resulting colonies, stored in PDA slants, were inoculated onto three-point agar plates and allowed to mature for a period of seven days. Detailed observations of colony morphology, including both obverse and reverse characteristics, were made on the PDA plates to facilitate preliminary identification of the fungal isolates at the genus level. To aid in taxonomic identification, lactocotton phenol blue staining was performed on the isolated samples, which were subsequently visualized under a light microscope (Labomed Lx 200, Mumbai, India) at 40× magnification. The arrangement of spores was examined to assist in taxonomic classification. Fungal isolates demonstrating potential for cypermethrin degradation were subjected to further identification through ITS sequencing [61].

Approximately 100 mg of mature fungal mycelium at 7 days of growth was disrupted using liquid nitrogen, and DNA was extracted from the powdered mycelium utilizing the NucleoSpin^®^ Plant II Kit (Macherey-Nagel, Allentown, PA, USA). The ITS gene was amplified using the universal primers ITS-1F (5′-TCCGTAGGTGAACCTGCGG-3′) and ITS-4R (5′-TCCTCCGCTTATTGATATGC-3′) in a thermal cycler (GeneAmp PCR System 9700, Applied Biosystems, Foster City, CA, USA). The quality of the DNA was assessed through agarose gel electrophoresis, followed by visualization under a UV transilluminator (Genei, Bangalore, India) and a gel documentation system (Bio-Rad, CA, USA). The amplified product was sequenced employing the BigDye Terminator v3.1 Cycle sequencing kit (Applied Biosystems, Foster City, CA, USA) and the ABI 3500 DNA analyzer (Applied Biosystems, Foster City, CA, USA). The obtained DNA sequence was compared against the ITS database in NCBI using BLAST to determine the fungal type and reference material [62]. Phylogenetic analysis was performed using ngphylogeny.fr, a Galaxy-based server, in Ala-carte mode [63]. The analysis involved a series of tools in the following sequence: Clustal Omega for sequence alignment, GBlocks for sequence cleaning, and MrBayes for tree construction. The resulting tree was saved in Newick format. Manual editing of the tree and its final presentation were accomplished using the Interactive Tree of Life (iTOL) website [64,65].

### 3.4. Screening for Resistance to Cypermethrin

The industrial grade cypermethrin dispersion (25% EC) was procured from reliable commercial suppliers located in Madurai, India. A concentrated solution of cypermethrin at a concentration of 1000 mg/L was prepared by diluting the commercial dispersion, and subsequent dilutions were made as per experimental requirements. To evaluate the resistance of fungal strains to cypermethrin, two nutrient media, namely potato dextrose agar (PDA) and minimal salt medium (MSM: KH_2_PO_4_:1.5 g/L, CaCl2·2H2O:0.01 g/L, Na_2_HPO_4_·12H_2_O:1.5 g/L, (NH_4_)_2_SO_4_:2.0 g/L, MgSO_4_·7H_2_O:0.2 g/L, FeSO_4_·7H_2_O:0.001 g/L, pH 7.0)), were supplemented with increasing concentrations of cypermethrin (CYP) ranging from 100, 200, 300, and 500 mg/L [8]. Fungal isolates that exhibited growth at a concentration of 500 mg/L of cypermethrin were selected for subsequent microcosm biodegradation studies.

### 3.5. Screening for Extracellular Enzymes Involved in Cypermethrin Degradation

Fungal isolates were subjected to plate-based assays to assess their ability to produce a range of extracellular enzymes, including protease, caseinase, cellulase, urease, esterase, and β-glucosidase. Additionally, the production of lignin-degrading enzymes such as manganese peroxidase and laccase was also investigated.

Caseinase: Fungal discs were aseptically placed on skimmed milk agar plates and incubated at room temperature for a period of 7 days. The presence of a distinct clear zone surrounding the fungal growth was indicative of positive caseinase production [66].

Cellulase: Agar plates of minimal salt medium supplemented with 500 mg/L CYP were formulated by incorporating 1% (*w*/*v*) microgranular cellulose. Fungal discs were aseptically placed on the plates, which were then incubated at room temperature for a period of 7 days. The development of a distinct clear zone around the fungal growth on the plates was regarded as a positive indication of cellulase production [66].

Esterase: Agar plates of minimal salt medium were formulated with the incorporation of bromocresol purple as a pH indicator at a specific pH value of 5.4. A separate solution of 10% tween was prepared and autoclaved before being added to the minimal salt medium agar in a volume ratio of 1:9. The fungal discs were aseptically placed on the plates and incubated at room temperature for a period of 7 days. The change in the color of the agar medium to purple was considered a positive indication of esterase production [66].

β-Glucosidase: The fungal isolates were inoculated onto sterile cellulose basal medium containing 0.5% esculin (*w*/*v*) and 2% ferric sulphate solution. Appearance of black colour in the plates after 5 days of incubation in darkness was considered to indicate the production of β-glucosidase [66].

Urease: The fungal isolates were aseptically inoculated onto a sterile urea agar-based medium. Following an incubation period of 8 days, the occurrence of a color change in the medium, specifically to pink or purple, was regarded as a positive outcome, indicating urease activity. Conversely, a color change to orange or yellow was considered a negative result, indicating the absence of urease activity [66].

Protease: The fungal isolates were aseptically inoculated onto sterile potato dextrose agar plates containing 1% gelatin as a substrate. Following incubation at ambient temperature for 48 h, the plates were subjected to flooding with an acidic mercuric chloride solution. The appearance of a clearance zone surrounding the fungal colonies was regarded as indicative of hydrolysis, suggesting the production of hydrolytic enzymes capable of gelatin degradation [66].

### 3.6. Qualitative Assay for Lignin-Degrading Enzymes

Staining Lignin Agar Method: Fungal strains were inoculated in MSM-containing ammonium tartarte (5 g/L) and 0.025% lignosulfonic acid. After incubation for up to 10 days, plates were flooded with freshly prepared 1% *w*/*v* FeCl_3_ and K_3_[Fe(CN)_6_]. Plates that showed clear zones around the colonies were an indication of oxidized phenolic compounds, while unoxidized phenols in undegraded lignin were found to turn blue-green [67].

Tannic acid agar method: Fungal discs were placed in MSM agar plates supplemented with 20% glucose (*w*/*v*) and 1% tannic acid (*w*/*v*) that were sterilized separately, incubated in darkness. The emergence of a brown oxidation zone around the fungal colonies represents the presence of polyphenol oxidase activity by fungal strains [67].

Bromophenol plate assay: Fungal strains were inoculated in a medium containing (*w*/*v*) 2% glucose, 3% malt extract, 1% agar, and bromophenol blue 0.02%. After 3 days of incubation, fungal colonies that decolorize bromophenol dye, produce clear halo zones around them [67].

Guaiacol agar plate assay: The isolated fungal colonies were plated on potato dextrose agar plates containing 0.01% guaiacol. After 8 days of incubation, plates were observed for the presence of brown-coloured halo zones surrounding the fungal strains, which was considered an indication of laccase production [67].

Methylene blue assay: The fungal discs were inoculated on potato dextrose agar plates containing 1 mM methylene blue as an indicator. After 3 days of incubation, fungal colonies were observed in clear zones around them, indicating the activity of the lignin peroxidase enzyme [67].

Phenol red assay: The fungal discs were inoculated on potato dextrose agar plates supplemented with 0.1 mg/mL of phenol. After 4 days of incubation, colonies with clear zones around them indicate manganese peroxidase enzyme activity [67].

### 3.7. Biodegradation of Cypermethrin

Cypermethrin biodegradation was studied in two different liquid nutrient media, potato dextrose broth (denoted as LNM to avoid confusion) and minimal salt medium (MSM). Cypermethrin biodegradation was compared between the fungal isolate PDB-B and a fungal consortium (C) consisting of strain A, PDB-B, J, UN2, M1 and SM108 (refer to Table 6). 

#### 3.7.1. Changes in pH and Biomass

The fungal strain PDB-B and fungal consortium comprising strains A, PDB-B, J, UN2, M1, and SM108 were cultivated in separate batches of potato dextrose broth (LNM) and minimal salt medium (MSM) supplemented with 500 mg/L of cypermethrin (CYP) and incubated for a duration of 24 days. Following incubation, 10 mL samples were recovered and subjected to centrifugation, resulting in the collection of a pellet. The pelletized biomass was subsequently dried in a hot air oven, and its weight was recorded. The pH of the supernatant was determined utilizing a pH meter, while the optical density of the fermented broth was measured at a wavelength of 700 nm using a colorimeter at regular intervals of 3 days over the course of 24 days [68].

#### 3.7.2. Microcosm Studies

The fungal strain PDB-B and the fungal consortium (C) consisting of strain A, PDB-B, J, UN2, M1, and SM108 were grown on potato dextrose agar for 7 days and 10 mm discs of mature mycelium were prepared. Five PDB-B discs and five discs of each fungal strain in the consortium were added to two batches of different liquid nutrient medium, 500 mL of potato dextrose broth (PDB) and minimal salt medium (MSM). They were also replicated as soil microcosm studies. 

For the study of biodegradation in soil microcosm, soil samples were sieved to remove debris, and autoclaved thrice to create a sterile environment. Subsequently, the soil samples were spiked with 500 mg/L of cypermethrin and inoculated with five discs of the mature fungal strain PDB-B and the fungal consortium (C). Likewise, control flasks without the addition of culture were kept for incubation for 24 days. A volume of 10 mL of sterile distilled water was added at regular intervals to moisten them. After the incubation period, soil samples were extracted for identification of residual metabolites by GC-MS analysis [69].

### 3.8. Sample Extraction and Analysis by GC-MS

The broth samples incubated for biodegradation were extracted using xylene and ethyl acetate added in equal volumes by separation funnel extraction (SFE). The eluent was evaporated and resuspended in acetonitrile. The metabolites in the soil samples were extracted with ethyl acetate using the Soxhlet apparatus and analysed using GC-MS (Shimadzu QP2020, Tokyo, Japan). The metabolites were matched against mass spectral libraries for compound identification [69].

### 3.9. Prediction and Data Processing of Transformation Products (TPs)

The microbial biodegradation pathway of cypermethrin, as well as its potential biotransformation products (TPs), were obtained and predicted utilizing the EAWAG Biodegradation/Biocatalysis Pathway (BBD) tool (http://eawag-bbd.ethz.ch, accessed on 8 March 2024) [70,71]. The structure of cypermethrin, represented by its Simplified Molecular Input Line Entry System (SMILES), was obtained from the PubChem database (https://pubchem.ncbi.nlm.nih.gov/#query=cypermethrin, accessed on 8 March 2024) and input into the EAWAG-BBD pathway prediction system to generate a list of potential TPs. The resulting TPs and their corresponding SMILES were compiled in a Microsoft Excel v. 16.83 file and manually compared with the data obtained from gas chromatography-mass spectrometry (GC-MS) analysis (Table 2 and Table 3).

### 3.10. Brine Shrimp Lethality Assay (BSLP)

The brine shrimp toxicity assay was used to assess the residual toxicity of biodegraded derivatives in soil samples as well as liquid batch samples [72]. *Artemia salina* cysts were incubated in a conical flask in seawater and hatched larvae were transferred to test tubes with 10 larvae each in triplicate. Cypermethrin extracts of samples from separatory funnel extractions were added to a tube containing 10 larvae each. Toxicity was determined by the mortality of *Artemia salina* and the percentage of lethality was calculated according to the equation:%Lethality = (No. of dead *Artemia salina*)/(Initial no. of *Artemia salina*) × 100

## 4. Conclusions

Fungal strains belonging to the *Aspergillus* genus capable of using cypermethrin as the sole carbon source were identified. The major strain, *Aspergillus terreus* PDB-B, was isolated from lake sediments and tolerates high concentrations of cypermethrin (up to 500 mg/L). It points to the existence of naturally occurring microbial communities with the potential to withstand and degrade high concentrations of environmental pollutants or the exposure of lake sediments to high concentrations of environmental pollutants. The biotransformation potential of *Aspergillus terreus* PDB-B was compared with that of a consortium consisting of multiple fungal strains, based on the results from plate-based assays. This shows that such translation from plate-based to in vitro assays and proceeding to soil microcosms is not possible always. Complex interactions between consortia, despite lack of antagonism between strains, can be intimidating to biodegradation. The study shows that the degradation process leads to the formation of (3-methylphenyl) methanol and isopropyl ether, among others, shedding light on potential degradation intermediates and end-products. The reduction in toxicity, as evidenced by the brine shrimp lethality assay, underscores the effectiveness of the fungal strains in not only breaking down cypermethrin but also in mitigating its environmental and health hazards.

## Figures and Tables

**Figure 1 molecules-29-01446-f001:**
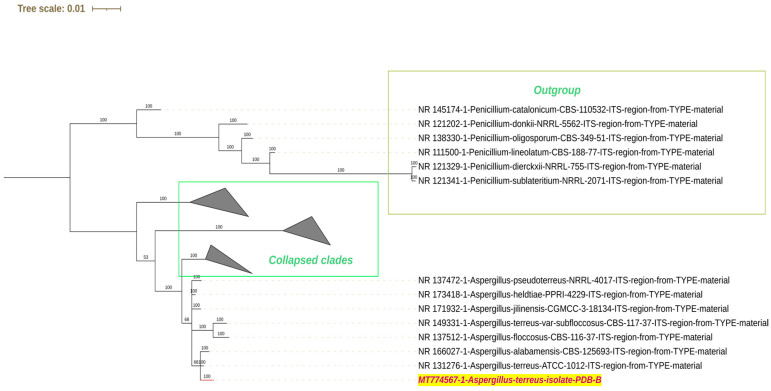
Phylogenetic tree-based identification of the fungal strain PDB-B from ITS sequencing. The strain used in the study is highlighted in red text.

**Figure 2 molecules-29-01446-f002:**
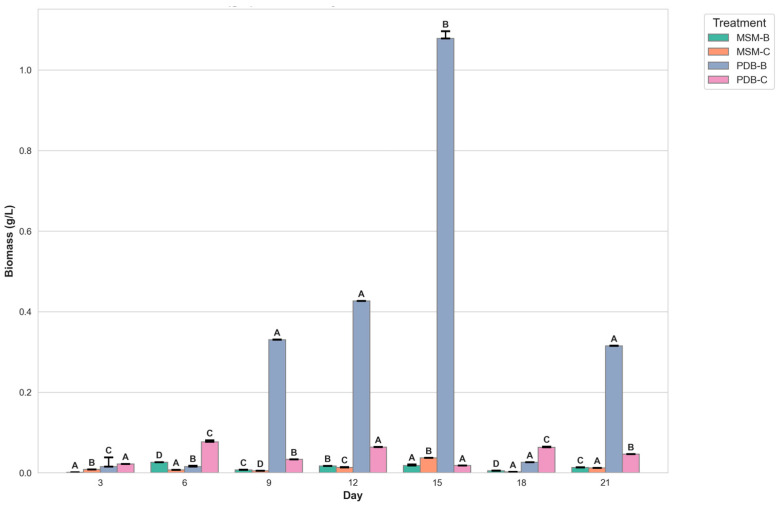
Graph showing the change in dry mycelial weight of fungi B and fungal consortium at 500 mg/L Cypermethrin (uppercase letters denotes statistical significance displayed using compact letter display).

**Figure 3 molecules-29-01446-f003:**
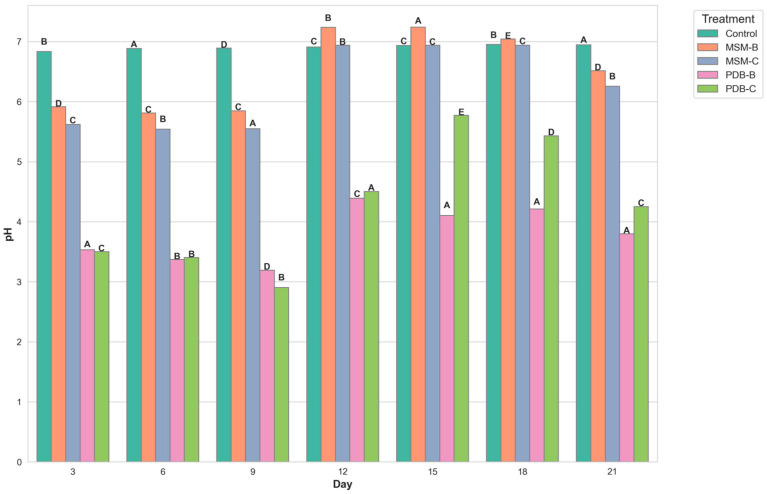
Graph showing the change in pH of fungal media inoculated with B and fungal consortium at 500 ppm Cypermethrin (uppercase letters denotes statistical significance displayed using compact letter display).

**Figure 4 molecules-29-01446-f004:**
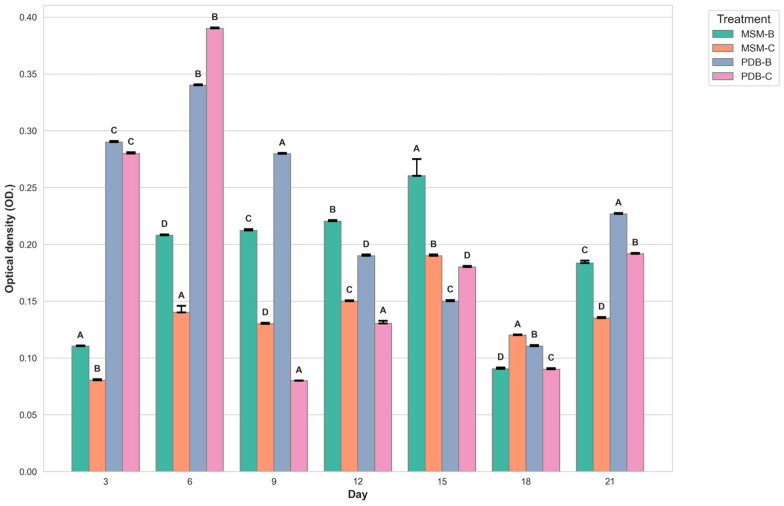
Graph showing the change in optical density of fungal media inoculated with B and fungal consortium at 500 ppm Cypermethrin. (uppercase letters denotes statistical significance displayed using compact letter display).

**Figure 5 molecules-29-01446-f005:**
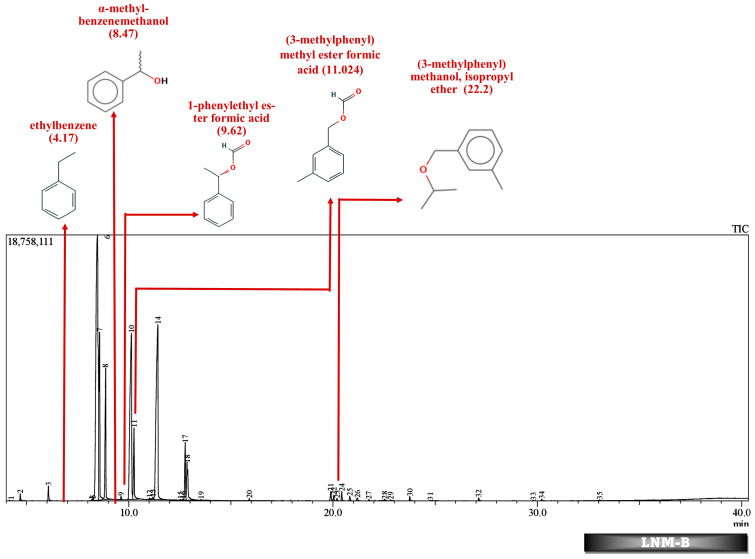
GC-MS chromatogram of liquid broth sample inoculated with 500 ppm Cypermethrin and fungal isolate B for 24 days.

**Figure 6 molecules-29-01446-f006:**
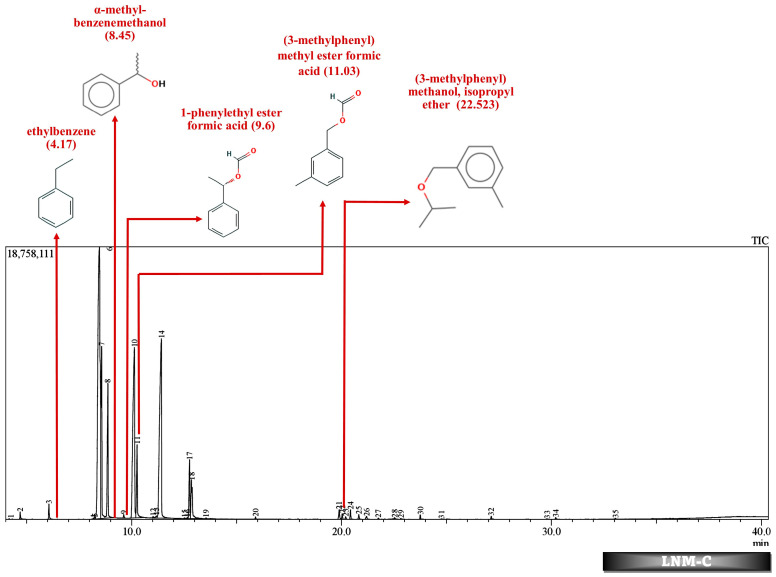
GC-MS chromatogram of liquid broth sample inoculated with 500 ppm Cypermethrin and fungal consortium C for 24 days.

**Figure 7 molecules-29-01446-f007:**
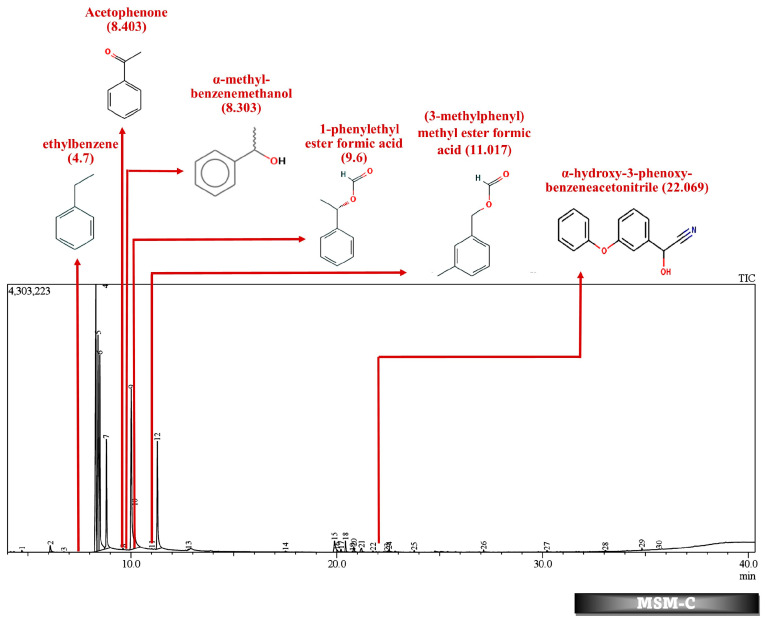
GC-MS chromatogram of MSM broth sample inoculated with 500 ppm Cypermethrin and fungal consortium C for 24 days.

**Figure 8 molecules-29-01446-f008:**
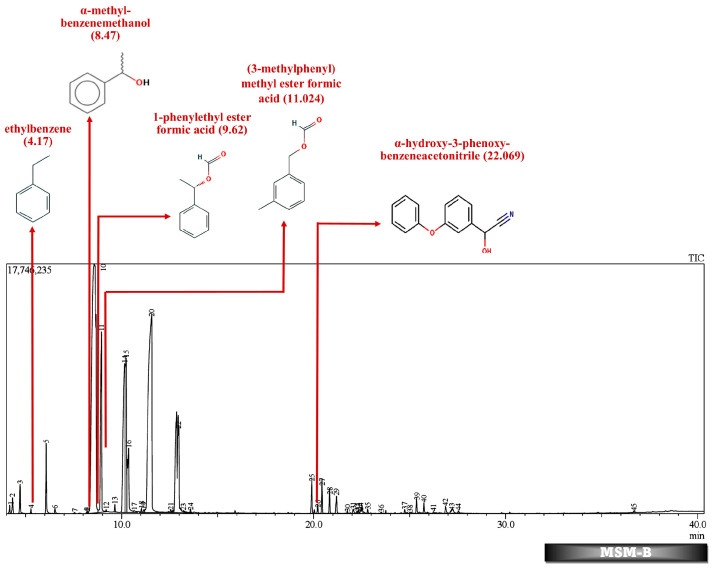
GC-MS chromatogram of MSM broth sample inoculated with 500 ppm Cypermethrin and fungal isolate B for 24 days.

**Figure 9 molecules-29-01446-f009:**
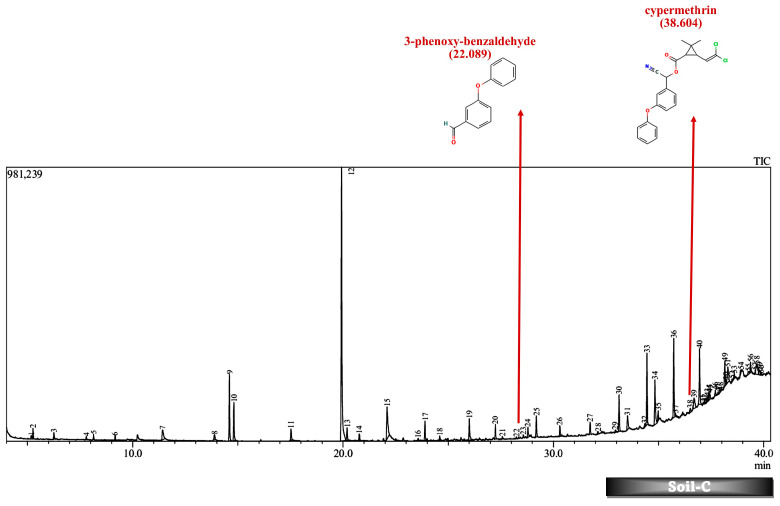
GC-MS chromatogram of soil microcosm inoculated with 500 ppm Cypermethrin and fungal consortium C for 24 days.

**Figure 10 molecules-29-01446-f010:**
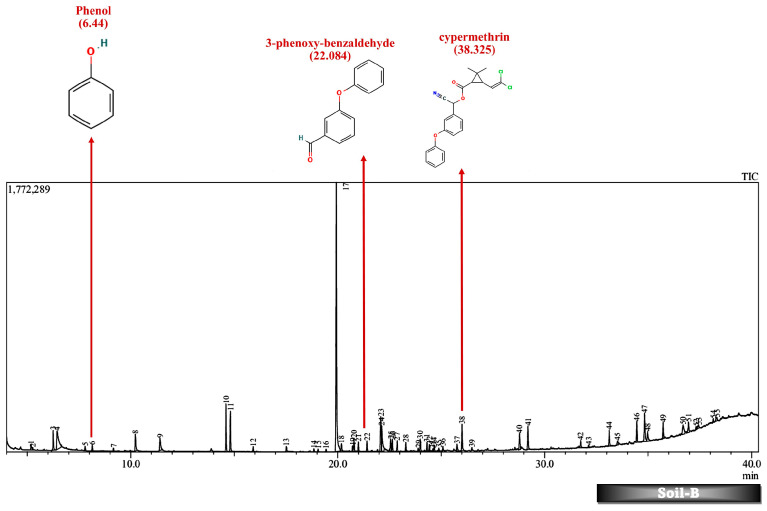
GC-MS chromatogram of soil microcosm inoculated with 500 ppm Cypermethrin and fungal isolate B for 24 days.

**Figure 11 molecules-29-01446-f011:**
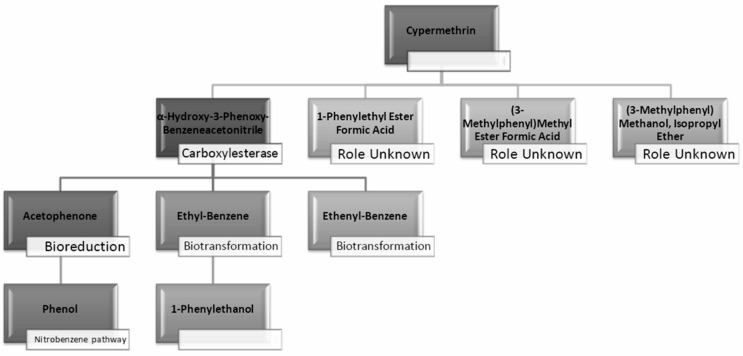
Microbial biodegradation pathway of cypermethrin predicted from GC-MS analysis of broth and soil samples (LNM-B, LNM-C, MSM-B, MSM-C, Soil-C and Soil-B). (Blank spaces indicate unidentified pathways/processes).

**Table 1 molecules-29-01446-t001:** Compilation of some microorganisms involved in the biodegradation of cypermethrin.

No.	Compound	Strain	Source of Isolate	Derived Metabolites	Conc. (mg/L)	% Degrad.	References
1.	Cypermethrin (technical grade)	*Bacillus thuringiensis* strain SG4 and strain SG2	Soil	3-phenoxybenzoic acid (3-PBA), 3-IAA, DCMU, Sulfosulfuron, Allethrin I	NA	91.3	[15]
2.	α-cypermethrin	*Acinetobacter schindleri*	grasshopper (*Poecilimon tauricola*)	3-phenoxybenzaldehyde, 3-PBA, phenol, muconic acid	100	68.4	[16]
3.	β-cypermethrin	*Bacillus cereus* GW-01	Sheep’s rumen chyme	3-PBA, phenol, catechol	100	∼60	[17]
4.	β-cypermethrin	*Lactobacillus pentosus* 3–27	β-CYP contaminated silage	3-PBA	50	96	[18]
5.	Cypermethrin (technical grade)	*Beauveria bassiana* (ITCC 913)	Indian Type Culture Collection (ITCC), India	4-hydroxybenzoate, cypermethrin, 2-(4-hydroxy phenoxy) benzoic acid methyl ester, 3,5,-dihydroxybenzoic acid, 3-(2,2-dichloroethenyl)-2,2-dimethylcyclopropane carboxylate, 3,5,-dimethoxy phenol, and phenol	0.12	NA *	[19]
6.	β-cypermethrin	*Bacillus cereus*	BT cotton cultivated and long-time pesticide-exposed soil	1-(2-acetoxyethyl)-3,6-diazahomoadamantan-9-one, benzene, 2 (dimethylamino)-, 1-ethyl-3-methyl, ethanethiol, silane, 1-(3-hydroxy-3-methylbutyl)-3, fumaric acid, 9-anthracenyltrimethyl-	100	NA *	[20]
7.	Cypermethrin (technical grade)	*Lysinibacillus cresolivuorans* his7	pesticide-contaminated soil	1H-purine-2,6-dione,3,7-dihydro 1,3,7 trimethy; benzene ethanamine, à-methyl-3-[4-methylphenyloxy]; 9-octadecenamide; 1,2-Benzenedicarboxylicacid-3-nitro; acetic acid (4-chloro-2-methylphenoxy)	2500	86.9	[21]
8.	β-cypermethrin	*Streptomyces toxytricini* D2	pesticide-exposed surface of cotton leaves	3-PBA, methyl salicylate, phthalic acid, phenol, and 3-phenoxy benzaldehyde	6% solution	80.71 ± 1.17	[14]
9.	α-cypermethrin (≥97%)	*Aspergillus* sp. PYR-P2	pesticide-contaminated soil	3-phenoxybenzaldehyde; α-cyano-3-phenoxybenzyl-3-(2,2-dichlorovinyl)-2,2-dimethyl cyclopropane carboxylate; 3-phenoxybenzoic acid	500	91.56	[22]
10.	Cypermethrin (technical grade)	*Bacillus thuringiensis* strain SG4	pesticide-contaminated soil from agricultural fields	3-phenoxybenzaldehyde; 2-hydroxy-2(3-phenoxyphenyl) acetonitrile; 3-(2,2-dichloroethenyl)-2,2-dimethyl cyclopropanecarboxylate; 2-hydroxy-2 (3-phenoxyphenyl) acetonitrile	50	80	[23]

* *NA denotes information unavailable in literature.*

**Table 2 molecules-29-01446-t002:** Heatmap of breakdown products of cypermethrin observed by GC-MS analysis.

CONTROL			LNM-C			LNM-B			MSM-B			MSM-C		
RT (min)	A%	Name	RT (min)	A%	Name	RT (min)	A%	Name	RT (min)	A%	Name	RT (min)	A%	Name
38.34	0.06	cypermethrin												
			22.523	0.08	(3-methylphenyl) methanol, isopropyl ether	22.2	0.11	(3-methylphenyl) methanol, isopropyl ether						
									22.069	0.17	α-hydroxy-3-phenoxy-benzeneacetonitrile			
			11.03	0.06	(3-methylphenyl) methyl ester formic acid	11.024	0.08	(3-methylphenyl) methyl ester formic acid	11.038	0.12	(3-methylphenyl) methyl ester formic acid	11.017	0.03	(3-methylphenyl) methyl ester formic acid
			9.6	0.13	1-phenylethyl ester formic acid	9.626	0.14	1-phenylethyl ester formic acid	9.647	0.14	1-phenylethyl ester formic acid	9.6	0.1	1-phenylethyl ester formic acid
			8.5	32.52	α-methyl-benzenemethanol	8.476	35.51	α-methyl-benzenemethanol	8.572	32.65	α-methyl-benzenemethanol	8.303	21.77	α-methyl-benzenemethanol
												8.403	19.56	1-phenyl-ethanone
									6.526	0.13	phenol			
			4.171	0.04	ethylbenzene	4.17	0.02	ethylbenzene	4.176	0.17	ethylbenzene	4.7	0.15	ethenylbenzene

*(Colored blanks indicates absence of corresponding compounds).*

**Table 3 molecules-29-01446-t003:** Breakdown Products of Cypermethrin in the Soil Microcosm Observed by GC-MS Analysis.

Soil-C			Soil-B		
Retention Time (min)	AREA%	Name	Retention Time (min)	AREA%	Name
38.604	0.44	cypermethrin	38.325	0.65	cypermethrin
			6.44	5.03	phenol
22.089	4.48	3-phenoxybenzaldehyde	22.084	4.46	3-phenoxybenzaldehyde

*(Colored blanks indicates absence of corresponding compounds).*

**Table 4 molecules-29-01446-t004:** List of transformation products (TPs) predicted from the EAWAG-BBD pathway and their SMILES.

SMILES	Name
CC1(C)C(C=C(Cl)Cl)C1C([O-)=O	3-(2,2-dichlorovinyl)-2,2-dimethylcyclopropanecarboxylate
OC(CN)c1ccc(Oc2cccc2)c1	2-hydroxy-2-(3-phenoxyphenyl)acetonitrile
CC1(CO)C(C=C(Cl)Cl)C1C([O-])=O	3-(2,2-dichloroethenyl)-2-(hydroxymethyl)-2-methylcyclopropane-1-carboxylic acid
OC(C([O-])=O)c1ccc(Oc2cccc2)c1	3-Phenoxy-4-Hydroxyphenylacetic Acid
O=Cc1cccc(Oc2cccc2)c1	3-phenoxybenzaldehyde
CC1(C)C(C=C(Cl)Cl)C1C(=O)OC(C([O-])=O)c1ccc(Oc2cccc2)c1	2-[3-(2,2-dichloroethenyl)-2,2-dimethylcyclopropanecarbonyl]oxy-2-(3-phenoxyphenyl)acetic acid
CC1(C=O)C(C=C(Cl)Cl)C1C([O-])=O	
OC(C([O-])=O)c1cccc(O)c1O	2-(2,3-dihydroxyphenyl)-2-hydroxyacetate
Oc1ccccc1	Phenol
OC(C([O-])=O)c1ccc(O)c(O)c1	2-(3,4-dihydroxyphenyl)-2-hydroxyacetate
OC(C([O-])=O)c1ccc(Oc2ccc(O)c2O)c1	methyl 3,5-dihydroxy-4phenylmethoxybenzoate
[O-]C(=O)C(=O)c1ccc(Oc2cccc2)c1	(3-acetylphenyl)benzoate
OC(C([O-])=O)c1cccc(O)c1	2-hydroxy-2-(3-hydroxyphenyl)acetate
Oc1ccccc1O	2-hydroxyphenoxy
[O-]C(=O)c1ccc(Oc2cccc2)c1	3-phenoxybenzoate

**Table 5 molecules-29-01446-t005:** Lethality rate of treated samples in BSLP assay.

Sample Name	Lethality (%) of *Artemia nauplii*
Control	100
Negative Control	73.33
Dilution between sample and ASW 5:5	
PDB-B	100
PDB-C	100
MSM-B	30
MSM-C	36.66
Dilution between sample and ASW 7:3	
PDB-B	100
PDB-C	100
MSM-B	30
MSM-C	33

**Table 6 molecules-29-01446-t006:** Representation of the respective sample names used in the study.

Sample Types	Media	Microorganisms	Sample Name
Liquid	Potato Dextrose Broth (PDB)	Fungi PDB-B	LNM-B
			LNM-C
	Mineral Salt Medium (MSM)	Fungal Consortium (A, PDB-B, J, UN2, M1, and SM108)	MSM-B
			MSM-C
Soil	-	Fungi PDB-B	Soil-B
	-	Fungal Consortium (A, PDB-B, J, UN2, M1, and SM108)	Soil-C

## Data Availability

Data are contained within the article and Appendix A.

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
