# Peer review of "Mycotransformation of Commercial Grade Cypermethrin Dispersion by Aspergillus terreus PDB-B Strain Isolated from Lake Sediments of Kulamangalam, Madurai"

_molecules, 2024, doi:10.3390/molecules29071446_

Round 1
Reviewer 1 Report
Comments and Suggestions for Authors
The authors in the manuscript entitled "Mycotransformation of commercial grade cypermethrin dispersion by Aspergillus terreus PDB-B strain isolated from lake sediments of Kulamangalam, Madurai” described the isolation of an Aspergillus terreus fungal strain from the sediments of a lake located in Madurai and the potential ability of this fungal strain to degrade a potentially dangerous pesticide known as cypermethrin. Although the interest of the paper from an environmental point of view could be relevant the experimental strategy plan by the authors is not well designed and contains some shortcomings that should be solved to reach the conclusions.
Major issues
1. The authors compared the activity of the PDB-B fungal strain versus the activity of a fungal consortium that also includes the PDB-strain. This is not a well comparison point. In addition, there is not any rationale explaining why they used a consortium of strains to compare.
2. Most of relevant results are included in supplementary material that is not accessible to check.
3. Some result sections are not support by figures or tables (2.6 e.g.)
4. One line of the conclusions is based on the lethality shrimp assay that is include in a supplementary table.
5. The experimental design of most of the assays should be improve. The controls used are not explain in most of the assays so maybe they were not used.
6. There is not statistical analysis of any of the results showed that support the conclusions obtained.
7. The quality of the figures should be improved due to not allow the correct understanding of them.
Author Response
Thank you for your remarks. Please see the attachment.

Reviewer 2 Report
Comments and Suggestions for Authors
Comments regarding the manuscript entitled “Mycotransformation of commercial grade cypermethrin dispersion by Aspergillus terreus PDB-B strain isolated from lake sediments of Kulamangalam, Madurai” (Manuscript ID: molecules-2856812).
1) The paper's subject is interesting because of its implications on health and environment.
2) In general, the manuscript is easy to follow but has several typographical errors. Please revise the manuscript.
3) Statistical treatment of data is missing.
4) Although the results are presented and discussed briefly and clearly, the authors should emphasize their significance and relevance.
5) Table 1, line 422, etc.: Make sure to italicize scientific names of organisms.
6) Lines 144-146 and Table S6: Please revise these results.
7) Figures 2 and 4: Please explain the behavior of the changes in dry mycelial weight and optical density for fungal isolate B and consortium cultures. Why the biomass increases and decreases over 21 days of incubation?
8) Filamentous fungi do not grow as single cells, but as hyphal filaments that cannot be quantified by the usual enumeration techniques used in bacteriology, such as optical density. Please analyze and discuss the optical density results. What is the relationship between dry weight and optical density?
9) What functional roles do consortium members play in cypermethrin biodegradation?
10) Authors should highlight the novelty of their research.
Comments on the Quality of English Language
The manuscript has several typographical errors.
Author Response

(The authors gave the same response as above.)

Round 2
Reviewer 2 Report
Comments and Suggestions for Authors
The authors took into account most of my comments.
Comments on the Quality of English LanguageThe quality of English language is fine.